# Variational Bayesian Reinforcement Learning with Regret Bounds

**Brendan O'Donoghue**
DeepMind, UK
`bodonoghue@google.com`

## Abstract

In reinforcement learning the Q-values summarize the expected future rewards that the agent will attain. However, they cannot capture the epistemic uncertainty about those rewards. In this work we derive a new Bellman operator with associated fixed point we call the 'knowledge values'. These K-values compress both the expected future rewards and the epistemic uncertainty into a single value, so that high uncertainty, high reward, or both, can yield high K-values. The key principle is to endow the agent with a risk-seeking utility function that is carefully tuned to balance exploration and exploitation. When the agent follows a Boltzmann policy over the K-values it yields a Bayes regret bound of $\tilde{O}(L^{3/2}\sqrt{SAT})$, where $L$ is the time horizon, $S$ is the number of states, $A$ is the number of actions, and $T$ is the total number of elapsed timesteps. We show deep connections of this approach to the soft-max and maximum-entropy strands of research in reinforcement learning.

## 1   Introduction and related work

In reinforcement learning (RL) an agent interacts with an environment in an episodic manner and attempts to maximize its return [54, 45]. In this work the environment is a Markov decision process (MDP) and we consider the Bayesian case where the agent has some prior information and as it gathers data it updates its posterior beliefs about the environment. In this setting the agent is faced with the choice of visiting well understood states or exploring the environment to determine the value of other states which might lead to a higher return. This trade-off is called the *exploration-exploitation* dilemma. One way to measure how well an agent balances this trade-off is a quantity called *regret*, which measures how sub-optimal the rewards the agent has received are so far, relative to the (unknown) optimal policy [12]. In the Bayesian case the natural quantity to consider is the Bayes regret, which is the expected regret under the agents prior information [17].

The optimal Bayesian policy can be formulated using *belief states*, but this is believed to be intractable for all but small problems [17]. Approximations to the optimal Bayesian policy exist, one of the most successful being Thompson sampling [53, 55] wherein the agent samples from the posterior over value functions and acts greedily with respect to that sample [38, 42, 27, 41]. It can be shown that this strategy yields both Bayesian and frequentist regret bounds under certain assumptions [3]. In practice, maintaining a posterior over value functions is intractable, and so instead the agent maintains the posterior over MDPs, and at each episode an MDP is sampled from this posterior, the value function for that sample is computed, and the policy acts greedily with respect to that value function. Due to the repeated sampling and computing of value functions this is practical only for small problems, though attempts have been made to extend it [37, 43].

Bayesian algorithms have the advantage of being able to incorporate prior information about the problem and, as we show in the numerical experiments, they tend to perform better than non-Bayesian approaches in practice [17, 51, 49]. Although typically Bayes regret bounds hold for any prior that satisfies the assumptions, the requirement that the prior over the MDP is known in advance

35th Conference on Neural Information Processing Systems (NeurIPS 2021).

is a disadvantage for Bayesian methods. One common concern is about performance degradation when the prior is misspecified. In this case it can be shown that the regret increases by a multiplicative factor related to the Radon-Nikodym derivative of the true prior with respect to the assumed prior [48, §3.1]. In other words, a Bayesian algorithm with sub-linear Bayes regret operating under a misspecified prior will still have sub-linear regret so long as the true prior is absolutely continuous with respect to the misspecified prior. Moreover, for any algorithm that satisfies a Bayes regret bound it is straightforward to derive a high-probability regret bound for any family of MDPs that has support under the prior, in a sense translating Bayes regret into frequentist regret; see [48, §3.1], [38, Appendix A] for details.

In this work we endow an agent with a particular *epistemic risk-seeking* utility function, where 'epistemic risk' refers to the Bayesian uncertainty that the agent has about the optimal value function of the MDP. In the context of RL, acting so as to maximize a risk-seeking utility function which assigns higher values to more uncertain actions is a form of *optimism in the face of uncertainty*, a well-known heuristic to encourage exploration [21, 5]. Any increasing convex function could be used as a risk-seeking utility, however, only the exponential utility function has a decomposition property which is required to derive a Bellman recursion [1, 44, 20, 46]. We call the fixed point of this Bellman operator the 'K-values' for *knowledge* since they compress the expected downstream reward and the downstream epistemic uncertainty at any state-action into a single quantity. A high K-value captures the fact that the state-action has a high expected Q-value or high uncertainty, or both. Following a Boltzmann policy over the K-values yields a practical algorithm that we call 'K-learning' which attains a Bayes regret upper bounded by $\tilde{O}(L^{3/2}\sqrt{SAT})$ where $L$ is the time horizon, $S$ is the number of states, $A$ is the number of actions per state, and $T$ is the total number of elapsed timesteps [11]. This regret bound matches the best known bound for Thompson sampling up to log factors [38] and is within a factor of $\sqrt{L}$ of the known information theoretic lower bound of $\Omega(L\sqrt{SAT})$ [23, Appendix D].

The update rule we derive is similar to that used in 'soft' Q-learning (so-called since the 'hard' max is replaced with a soft-max) [6, 16, 18, 30, 47]. These approaches are very closely related to maximum entropy reinforcement learning techniques which add an entropy regularization 'bonus' to prevent early convergence to deterministic policies and thereby heuristically encourage exploration [57, 59, 28, 35, 2, 25]. In our work the soft-max operator and entropy regularization arise naturally from the view of the agent as maximizing a risk-seeking exponential utility. Furthermore, in contrast to these other approaches, the entropy regularization is not a fixed hyper-parameter but something we explicitly control (or optimize for) in order to carefully trade-off exploration and exploitation.

The algorithm we derive in this work is model-based, *i.e.*, requires estimating the full transition function for each state. There is a parallel strand of work deriving regret and complexity bounds for *model-free* algorithms, primarily based on extensions of Q-learning [23, 58, 26]. We do not make a detailed comparison between the two approaches here other than to highlight the advantage that model-free algorithms have both in terms of storage and in computational requirements. On the other hand, in the numerical experiments model-based approaches tend to outperform the model-free algorithms. We conjecture that an online, model-free version of K-learning with similar regret guarantees can be derived using tools developed by the model-free community. We leave exploring this to future work.

## 1.1 Summary of main results

- We consider an agent endowed an epistemic risk-seeking utility function and derive a new optimistic Bellman operator that incorporates the 'value' from epistemic uncertainty about the MDP. The new operator replaces the usual max operator with a soft-max and it incorporates a 'bonus' that depends on state-action visitation. In the limit of zero uncertainty the new operator reduces to the standard optimal Bellman operator.

- At each episode we solve the optimistic Bellman equation for the 'K-values' which represent the utility of a particular state and action. If the agent follows a Boltzmann policy over the K-values with a carefully chosen temperature schedule then it will enjoy a sub-linear Bayes regret bound.

- To the best of our knowledge this is the first work to show that soft-max operators and maximum entropy policies in RL can provably yield good performance as measured by Bayes regret. Similarly, we believe this is the first result deriving a Bayes regret bound for

a Boltzmann policy in RL. This puts maximum entropy, soft-max operators, and Boltzmann exploration in a principled Bayesian context and shows that they are naturally derived from endowing the agent with an exponential utility function.

## 2 Markov decision processes

In a Markov decision process (MDP) an agent interacts with an environment in a series of episodes and attempts to maximize the cumulative rewards. A finite horizon, discrete-time, discrete MDP is given by the tuple $\mathcal{M} = \{\mathcal{S}, \mathcal{A}, R, P, L, \rho\}$, where $\mathcal{S} = \{1, \ldots, S\}$ is the state-space, $\mathcal{A} = \{1, \ldots, A\}$ is the action-space, $R_l(s, a)$ is a probability distribution over the rewards received by the agent at state $s$ taking action $a$ at timestep $l$, $P_l(s' \mid s, a) \in [0, 1]$ is the probability the agent will transition to state $s'$ after taking action $a$ in state $s$ at timestep $l$, $L \in \mathbb{N}$ is the episode length, and $\rho$ is the initial state distribution. Concretely, the initial state $s \in \mathcal{S}$ of the agent is sampled from $\rho$, then for timesteps $l = 1, \ldots, L$ the agent is in state $s \in \mathcal{S}$, selects action $a \in \mathcal{A}$, receives reward $r \sim R_l(s, a)$ with mean $\mu_l(s, a) \in \mathbb{R}$ and transitions to the next state $s'$ with probability $P_l(s' \mid s, a)$. After timestep $L$ the episode terminates and the state is reset. We assume that at the beginning of learning the agent does not know the reward or transition probabilities and must learn about them by interacting with the environment. We consider the Bayesian case in which the mean reward $\mu$ and the transition probabilities $P$ are sampled from a known prior $\phi$. We assume that the agent knows $S$, $A$, $L$, and the reward noise distribution.

An agent following policy $\pi_l : \mathcal{S} \times \mathcal{A} \to [0, 1]$ at state $s \in \mathcal{S}$ at time $l$ selects action $a$ with probability $\pi_l(s, a)$. The Bellman equation relates the value of actions taken at the current timestep to future returns through the *Q-values* and the associated *value function* [8], which for policy $\pi$ are denoted $Q_l^\pi \in \mathbb{R}^{S \times A}$ and $V_l^\pi \in \mathbb{R}^S$ for $l = 1, \ldots, L + 1$, and satisfy

$$Q_l^\pi = \mathcal{T}_l^\pi Q_{l+1}^\pi, \quad V_l^\pi(s) = \sum_{a \in \mathcal{A}} \pi_l(s, a) Q_l^\pi(s, a), \tag{1}$$

for $l = 1, \ldots, L$ where $Q_{L+1} \equiv 0$ and where the Bellman operator for policy $\pi$ at step $l$ is defined as

$$(\mathcal{T}_l^\pi Q_{l+1}^\pi)(s, a) := \mu_l(s, a) + \sum_{s' \in \mathcal{S}} P_l(s' \mid s, a) \sum_{a' \in \mathcal{A}} \pi_l(s', a') Q_{l+1}^\pi(s', a').$$

The expected performance of policy $\pi$ is denoted $J^\pi = \mathbb{E}_{s \sim \rho} V_1^\pi(s)$. An optimal policy satisfies $\pi^\star \in \operatorname{argmax}_\pi J^\pi$ and induces associated optimal Q-values and value function given by

$$Q_l^\star = \mathcal{T}_l^\star Q_{l+1}^\star, \quad V_l^\star(s) = \max_a Q_l^\star(s, a). \tag{2}$$

for $l = 1, \ldots, L$, where $Q_{L+1}^\star \equiv 0$ and where the optimal Bellman operator is defined at step $l$ as

$$(\mathcal{T}_l^\star Q_{l+1}^\star)(s, a) := \mu_l(s, a) + \sum_{s' \in \mathcal{S}} P_l(s' \mid s, a) \max_{a'} Q_{l+1}^\star(s', a'). \tag{3}$$

### 2.1 Regret

If the mean reward $\mu$ and transition function $P$ are known exactly then (in principle) we could solve (2) via dynamic programming [9]. However, in practice these are not known and so the agent must gather data by interacting with the environment over a series of episodes. The key trade-off is the *exploration-exploitation* dilemma, whereby an agent must take possibly suboptimal actions in order to learn about the MDP. Here we are interested in the *regret* up to time $T$, which is how sub-optimal the agent's policy has been so far. The regret for an algorithm producing policies $\pi^t$, $t = 1, \ldots, N$ executing on MDP $\mathcal{M}$ is defined as

$$\mathcal{R}_\mathcal{M}(T) := \sum_{t=1}^N \mathbb{E}_{s \sim \rho}(V_1^\star(s) - V_1^{\pi^t}(s)),$$

where $N := \lceil T/L \rceil$ is the number of elapsed episodes. In this manuscript we take the case where $\mathcal{M}$ is sampled from a known prior $\phi$ and we want to minimize the expected regret of our algorithm under that prior distribution. This is referred to as the *Bayes regret*:

$$\mathcal{BR}_\phi(T) := \mathbb{E}_{\mathcal{M} \sim \phi} \mathcal{R}_\mathcal{M}(T) = \mathbb{E}_{\mathcal{M} \sim \phi} \sum_{t=1}^N \mathbb{E}_{s \sim \rho}(V_1^\star(s) - V_1^{\pi^t}(s)). \tag{4}$$

In the Bayesian view of the RL problem the quantities $\mu$ and $P$ are random variables, and consequently the optimal Q-values $Q^\star$, policy $\pi^\star$, and value function $V^\star$ are also random variables that must be learned about by gathering data from the environment. We shall denote by $\mathcal{F}_t$ the sigma-algebra generated by all the history *before* episode $t$ where $\mathcal{F}_1 = \emptyset$ and we shall use $\mathbb{E}^t$ to denote $\mathbb{E}(\,\cdot\mid\mathcal{F}_t)$, the expectation conditioned on $\mathcal{F}_t$. For example, with this notation $\mathbb{E}^t Q^\star$ denotes the expected optimal Q-values under the posterior before the start of episode $t$.

## 3  K-learning

Now we present Knowledge Learning (K-learning), a Bayesian RL algorithm that satisfies a sublinear Bayes regret guarantee. In standard dynamic programming the Q-values are the unique fixed point of the Bellman equation, and they summarize the expected future reward when following a particular policy. However, standard Q-learning is not able to incorporate any of the uncertainty about future rewards or transitions. In this work we develop a new Bellman operator with associated fixed point we call the 'K-values' which represent *both* the expected future rewards and the uncertainty about those rewards. These two quantities are compressed into a single value by the use of an exponential risk-seeking utility function, which is tuned to trade-off exploration and exploitation. In this section we develop the intuition behind the approach and defer all proofs to the appendix. We begin with the main assumption that we require for the analysis (this assumption is standard, see, *e.g.*, [41]).

**Assumption 1.** *The mean rewards are bounded in $[0,1]$ almost surely with independent priors, the reward noise is additive $\sigma$-sub-Gaussian, and the prior over transition functions is independent Dirichlet.*

### 3.1  Utility functions and the certainty equivalent value

A utility function $u : \mathbb{R} \to \mathbb{R}$ measures an agents preferences over outcomes [56]. If $u(x) > u(y)$ for some $x, y \in \mathbb{R}$ then the agent prefers $x$ to $y$, since it derives more utility from $x$ than from $y$. If $u$ is convex then it is referred to as *risk-seeking*, since $\mathbb{E}u(X) \geq u(\mathbb{E}X)$ for random variable $X$ due to Jensen's inequality. The particular utility function we shall use is the exponential utility $u(x) := \tau(\exp(x/\tau) - 1)$ for some $\tau \geq 0$. The *certainty equivalent value* of a random variable under utility $u$ measures how much guaranteed payoff is equivalent to a random payoff, and for $Q_l^\star(s,a)$ under the exponential utility is given by

$$\mathcal{Q}_l^t(s,a) := u^{-1}(\mathbb{E}^t u(Q_l^\star(s,a)) = \tau \log \mathbb{E}^t \exp(Q_l^\star(s,a)/\tau). \tag{5}$$

This is the key quantity we use to summarize the expected value and the epistemic uncertainty into a single value. As an example, consider a stochastic multi-armed bandit (*i.e.*, an MDP with $L = 1$ and $S = 1$) where the prior over the rewards and the reward noise are independent Gaussian distributions. At round $t$ the posterior over $Q^\star(a)$ is given by $\mathcal{N}(\mu_a^t, (\sigma_a^t)^2)$ for some $\mu_a^t$ and $\sigma_a^t$ for each action $a$, due to the conjugacy of the prior and the likelihood. In this case the certainty equivalent value can be calculated using the Gaussian cumulant generating function, and is given by $\mathcal{Q}^t(a) = \mu_a^t + (1/2)(\sigma_a^t)^2/\tau_t$. Evidently, this value is combining the expected reward and the epistemic uncertainty into a single quantity with $\tau_t$ controlling the trade-off, and the value is higher for arms with more epistemic uncertainty. Now consider the policy $\pi^t(a) \propto \exp(\mathcal{Q}^t(a)/\tau_t)$. This policy will in general assign greater probability to more uncertain actions, *i.e.*, the policy is *optimistic*. We shall show later that for a carefully selected sequence of temperatures $\tau_t$ we can ensure that this policy enjoys a $\tilde{O}(\sqrt{AT})$ Bayes regret bound for this bandit case. In the more general RL case the posterior over the Q-values is a complicated function of downstream uncertainties and is not a simple distribution like a Gaussian, but the intuition is the same.

The choice of the exponential utility may seem arbitrary, but in fact it is the unique utility function that has the property that the certainty equivalent value of the sum of two independent random variables is equal to the sum of their certainty equivalent values [1, 44, 20, 46]. This property is crucial for deriving a Bellman recursion, which is necessary for dynamic programming to be applicable.

## 3.2 Optimistic Bellman operator

A risk-seeking agent would compute the certainty equivalent value of the Q-values under the endowed utility function and then act to maximize this value. However, computing the certainty equivalent values in a full MDP is challenging. The main result (proved in the appendix) is that $\mathcal{Q}^t$ satisfies a Bellman *inequality* with a particular optimistic (*i.e.*, risk-seeking) Bellman operator, which for episode $t$ and timestep $l$ is given by

$$\mathcal{B}_l^t(\tau, K_l)(s,a) = \mathbb{E}^t \mu_l(s,a) + \frac{\sigma^2 + (L-l)^2}{2\tau(n_l^t(s,a) \vee 1)} + \sum_{s' \in \mathcal{S}} \mathbb{E}^t P_l(s' \mid s,a)(\tau \log \sum_{a' \in \mathcal{A}} \exp(K_l(s',a')/\tau)) \tag{6}$$

for inputs $\tau \geq 0$, $K_l \in \mathbb{R}^{S \times A}$ where $n_l^t(s,a)$ is the visitation count of the agent to state-action $(s,a)$ at timestep $l$ before episode $t$ and $(\cdot \vee 1) \coloneqq \max(\cdot, 1)$. Concretely we have that for any $(s,a) \in \mathcal{S} \times \mathcal{A}$

$$\mathcal{Q}_l^t(s,a) \leq \mathcal{B}_l^t(\tau, \mathcal{Q}_{l+1}^t)(s,a), \quad l = 1, \dots, L.$$

From this fact we show that the fixed point of the optimistic Bellman operator yields a guaranteed upper bound on $\mathcal{Q}^t$, *i.e.*,

$$\left( K_l^t = \mathcal{B}_l^t(\tau, K_{l+1}^t), \ l = 1, \dots, L \right) \Rightarrow \left( K_l^t \geq \mathcal{Q}_l^t, \ l = 1, \dots, L \right). \tag{7}$$

We refer to the fixed point as the 'K-values' (for knowledge) and we shall show that they are a sufficiently faithful approximation of $\mathcal{Q}^t$ to provide a Bayes regret guarantee when used instead of the certainty equivalent values in a policy.

Let us compare the optimistic Bellman operator $\mathcal{B}^t$ to the optimal Bellman operator $\mathcal{T}^\star$ defined in (3). The first difference is that the random variables $\mu$ and $P$ are replaced with their expectation under the posterior; in $\mathcal{T}^\star$ they are assumed to be known. Secondly, the rewards in the optimistic Bellman operator have been augmented with a bonus that depends on the visitation counts $n^t$. This bonus encourages the agent to visit state-actions that have been visited less frequently. Finally, the hard-max of the optimal Bellman operator has been replaced with a soft-max. Note that in the limit of zero uncertainty in the MDP (take $n_l^t(s,a) \to \infty$ for all $(s,a)$) we have $\mathcal{B}_l^t(0,\cdot) = \mathcal{T}_l^\star$ and we recover the optimal Bellman operator, and consequently in that case $K_l^t(s,a) = \mathcal{Q}_l^t(s,a) = Q_l^\star(s,a)$. In other words, the optimistic Bellman operator and associated K-values generalize the optimal Bellman operator and optimal Q-values to the epistemically uncertain case, and in the limit of zero uncertainty we recover the optimal quantities.

## 3.3 Maximum entropy policy

An agent that acts to maximize its K-values is (approximately) acting to maximize its risk-seeking utility. In the appendix we show that the policy that maximizes the expected K-values with *entropy regularization* is the natural policy to use, which is motivated by the variational description of the soft-max

$$\max_{\pi_l(s) \in \Delta_A} \left( \sum_{a \in \mathcal{A}} \pi_l(s,a) K_l^t(s,a) + \tau_t H(\pi_l(s)) \right) = \tau_t \log \sum_{a \in \mathcal{A}} \exp(K_l^t(s,a)/\tau_t) \tag{8}$$

where $\Delta_A$ is the probability simplex of dimension $A - 1$ and $H$ is entropy, *i.e.*, $H(\pi_l(s)) = -\sum_{a \in \mathcal{A}} \pi_l(s,a) \log \pi_l(s,a)$ [13]. The maximum is achieved by the Boltzmann (or Gibbs) distribution with temperature $\tau_t$

$$\pi_l^t(s,a) \propto \exp(K_l^t(s,a)/\tau_t). \tag{9}$$

This variational principle also arises in statistical mechanics where Eq. (8) refers to the negative Helmholtz free energy and the distribution in Eq. (9) describes the probability that the system at temperature $\tau_t$ is in a particular 'state' [22].

## 3.4 Choosing the risk-seeking parameter / temperature

The optimistic Bellman operator, K-values, and associated policy depend on the parameter $\tau_t$. By carefully controlling this parameter we ensure that the agent balances exploration and exploitation.

---
**Algorithm 1** K-learning for episodic MDPs
---

   **Input:** MDP $\mathcal{M} = \{\mathcal{S}, \mathcal{A}, R, P, L, \rho\}$,
   **for** episode $t = 1, 2, \ldots$ **do**
       calculate $\tau_t$ using (10) or (11)
       set $K_{L+1}^t \equiv 0$
       compute $K_l^t = \mathcal{B}_l^t(\tau_t, K_{l+1}^t)$, for $l = L, \ldots, 1$, using (6)
       execute policy $\pi_l^t(s, a) \propto \exp(K_l^t(s, a)/\tau_t)$, for $l = 1, \ldots, L$
   **end for**
---

We present two ways to do so, the first of which is to follow the schedule

$$\tau_t = \sqrt{\frac{(\sigma^2 + L^2)SA(1 + \log t)}{4t \log A}}. \tag{10}$$

Alternatively, we find the $\tau_t$ that yields the tightest bound in the maximal inequality in (13). This turns out to be a convex optimization problem

$$
\begin{aligned}
\text{minimize} \quad & \mathbb{E}_{s \sim \rho} \left( \tau \log \sum_{a \in \mathcal{A}} \exp(K_1(s, a)/\tau) \right) \\
\text{subject to} \quad & K_l \geq \mathcal{B}_l^t(\tau, K_{l+1}), \quad l = 1, \ldots, L, \\
& K_{L+1} \equiv 0,
\end{aligned}
\tag{11}
$$

with variables $\tau \geq 0$ and $K_l \in \mathbb{R}^{S \times A}$, $l = 1, \ldots L + 1$. This is convex jointly in $\tau$ and $K$ since the Bellman operator is convex in both arguments and the perspective of the soft-max term in the objective is convex [10]. This generalizes the linear programming formulation of dynamic programming to the case where we have uncertainty over the parameters that define the MDP [45, 9]. Problem (11) is an *exponential cone program* and can be solved efficiently using modern methods [32, 33, 15, 50, 31].

Both of these schemes for choosing $\tau_t$ yield a Bayes regret bound, though in practice the $\tau_t$ obtained by solving (11) tends to perform better. Note that since actions are sampled from the stochastic policy in Eq. (9) we refer to K-learning a *randomized* strategy. If K-learning is run with the optimal choice of temperature $\tau_t^\star$ then it is additionally a *stationary* strategy in that the action distribution depends solely on the posteriors and is otherwise independent of the time period [49].

### 3.5 Regret analysis

**Theorem 1.** *Under assumption 1 the K-learning algorithm 1 satisfies Bayes regret bound*

$$
\begin{aligned}
\mathcal{BR}_\phi(T) &\leq 2\sqrt{(\sigma^2 + L^2)LSAT \log A(1 + \log T/L)} \\
&= \tilde{O}(L^{3/2}\sqrt{SAT}).
\end{aligned}
\tag{12}
$$

The full proof is included in the appendix. The main challenge is showing that the certainty equivalent values $\mathcal{Q}^t$ satisfy the Bellman inequality with the optimistic Bellman operator (6). This is used to show that the K-values upper bound $\mathcal{Q}^t$ (Eq. (7)) from which we derive the following maximal inequality

$$\mathbb{E}^t \max_a Q_l^\star(s, a) \leq \tau_t \log \sum_{a \in \mathcal{A}} \exp(\mathcal{Q}_l^t(s, a)/\tau_t) \leq \tau_t \log \sum_{a \in \mathcal{A}} \exp(K_l^t(s, a)/\tau_t). \tag{13}$$

From this and the Bellman recursions that the K-values and the Q-values must satisfy, we can 'unroll' the Bayes regret (4) over the MDP. Using the variational description of the soft-max in Eq. (8) we can cancel out the expected reward terms leaving us with a sum over 'uncertainty' terms. Since the uncertainty is reduced by the agent visiting uncertain states we can bound the remaining terms using a standard pigeon-hole argument. Finally, the temperature $\tau_t$ is a free-parameter for each episode $t$, so we can choose it so as to minimize the upper bound. This yields the final result.

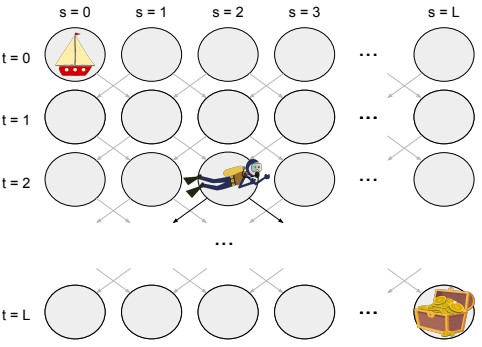

Figure 1: The DeepSea MDP

The Bayes regret bound in the above theorem matches the best known bound for Thompson sampling up to log factors (once the regret bound from [38] is translated into the slightly more general case we consider here where the mean reward and transition function can differ for each timestep $l = 1, \ldots, L$). Moreover, the above regret bound is within a factor of $\sqrt{L}$ of the known information theoretic lower bound of $\mathcal{R}(T) \geq \Omega(L\sqrt{SAT})$ [23, Appendix D].

Intuitively speaking, K-values are higher where the agent has high epistemic uncertainty. Higher K-values will make the agent more likely to take the actions that lead to those states. Over time states with high uncertainty will be visited and the uncertainty about them will be resolved. The temperature parameter $\tau_t$ is controlling the balance between exploration and exploitation.

## 4   Numerical experiments

In this section we compare the performance of both the temperature scheduled and optimized temperature variants of K-learning against several other methods in the literature. We consider a small tabular MDP called *DeepSea* [39] shown in Figure 1, which can be thought of as an unrolled version of the RiverSwim environment [52]. This MDP can be visualized as an $L \times L$ grid where the agent starts at the top row and leftmost column. At each time-period the agent can move left or right and descends one row. The only positive reward is at the bottom right cell. In order to reach this cell the agent must take the 'right' action every timestep. After choosing action 'left' the agent receives a random reward with zero mean, and after choosing right the agent receives a random reward with small negative mean. At the bottom rightmost corner of the grid the agent receives a random reward with mean one. Although this is a toy example it provides a challenging 'needle in a haystack' exploration problem. Any algorithm that does exploration via a simple heuristic like local dithering will take time exponential in the depth $L$ to reach the goal. Policies that perform deep exploration can learn much faster [38, 43].

In Figure 2 we show the time required to 'solve' the problem as a function of the depth of the environment, averaged over $5$ seeds for each algorithm. We define 'time to solve' to be the first episode at which the agent has reached the rewarding state in at least $10\%$ of the episodes so far. If an agent fails to solve an instance within $10^5$ episodes we do not plot that point, which is why some of the traces appear to abruptly stop. We compare two dithering approaches, Q-learning with epsilon-greedy ($\epsilon = 0.1$) and soft-Q-learning [18] ($\tau = 0.05$), against principled exploration strategies RLSVI [39], UCBVI [7], optimistic Q-learning (OQL) [23], BEB [24], Thompson sampling [38] and two variants of K-learning, one using the $\tau_t$ schedule (10) and the other using the optimal choice $\tau_t^\star$ from solving (11). Soft Q-learning is similar to K-learning with two major differences: the temperature term is a fixed hyperparameter and there is no optimism bonus added to the rewards. These differences prevent soft Q-learning from satisfying a regret bound and typically it cannot solve difficult exploration tasks in general [36]. We also ran comparisons against BOLT [4], UCFH [14],

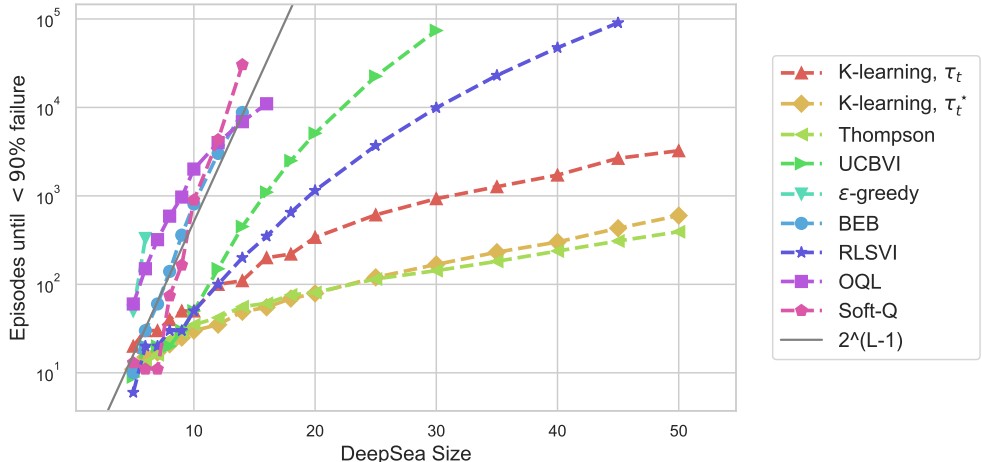

Figure 2: Learning time on DeepSea.

and UCRL2 [21] but they did not perform much better than the dithering approaches and contributed to the clutter in the figure so we do not show them.

As expected, the two 'dithering' approaches are unable to handle the problem as the depth exceeds a small value; they fail to solve the problem within $10^5$ episodes for problems larger than $L = 6$ for epsilon-greedy and $L = 14$ for soft-Q-learning. These approaches are taking time exponential in the size of the problem to solve the problem, which is seen by comparing their performance to the grey dashed line which plots $2^{L-1}$. The other approaches scale more gracefully, however clearly Thompson sampling and K-learning are the most efficient. The optimal choice of K-learning appears to perform slightly better than the scheduled temperature variant, which is unsurprising since it is derived from a tighter upper bound on the regret.

In Figures 3 and 4 we show the progress of K-learning using $\tau_t^\star$ and Soft Q-learning for a single seed running on the $L = 50$ depth DeepSea environment. In the top row of each figure we show the value of each state over time, defined for K-learning as

$$\tilde{V}_l^t(s) = \tau_t \log \sum_{a \in \mathcal{A}} \exp(K_l^t(s, a)/\tau_t), \tag{14}$$

and analogously for Soft Q-learning. The bottom row shows the log of the visitation counts over time. Although both approaches start similarly, they quickly diverge in their behavior. If we examine the K-learning plots, it is clear that the agent is visiting more of the bottom row as the episodes proceed. This is driven by the fact that the value, which incorporates the epistemic uncertainty, is high for the unvisited states. Concretely, take the $t = 300$ case; at this point the K-learning agent has not yet visited the rewarding bottom right state, but the value is very high for that region of the state space and shortly after this it reaches the reward for the first time. By the $t = 1000$ plot the agent is travelling along the diagonal to gather the reward consistently. Contrast this to Soft Q-learning, where the agent does not make it even halfway across the grid after $t = 1000$ episodes. This is because the soft Q-values do not capture any uncertainty about the environment, so the agent has no incentive to explore and visit new states. The only exploration that soft Q-learning is performing is merely the local dithering arising from using a Boltzmann policy with a nonzero temperature. Indeed, the soft value function barely changes in this case since the agent is consistently gathering zero-mean rewards; any fluctuation in the value function arises merely from the noise in the random rewards.

## 5 Conclusions

In this work we endowed a reinforcement learning agent with a *risk-seeking* utility, which encourages the agent to take actions that lead to less epistemically certain states. This yields a Bayesian

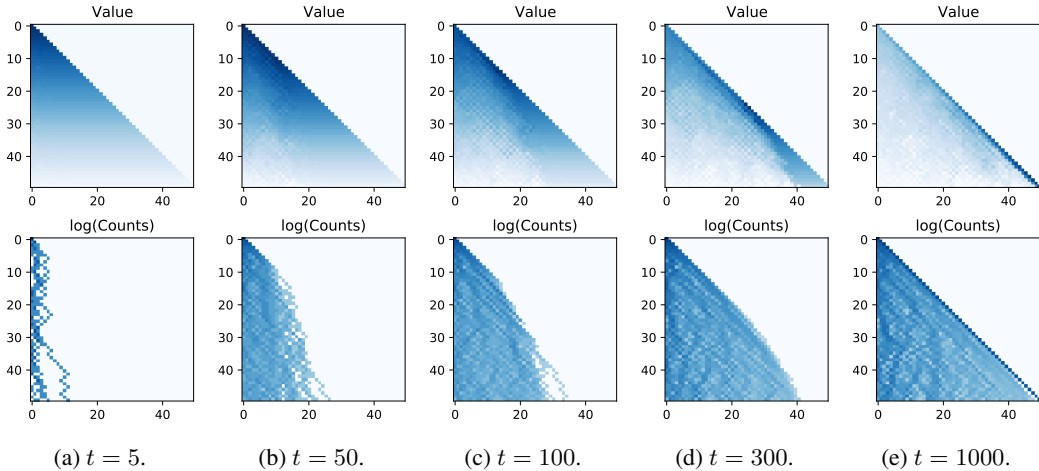

(a) $t = 5$.      (b) $t = 50$.      (c) $t = 100$.      (d) $t = 300$.      (e) $t = 1000$.

Figure 3: Value (as defined in (14)) and log visitation count for each state in DeepSea under K-learning using $\tau_t^\star$. Darker color indicates larger value.

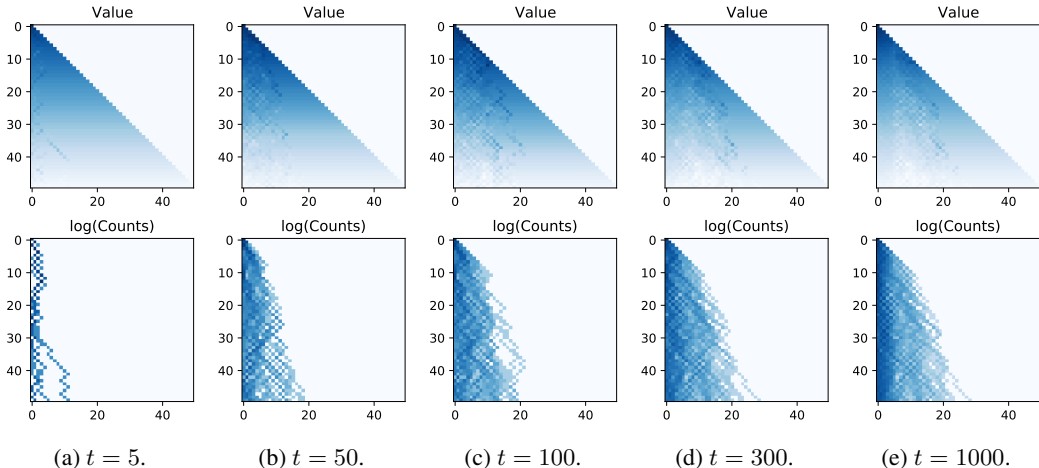

(a) $t = 5$.      (b) $t = 50$.      (c) $t = 100$.      (d) $t = 300$.      (e) $t = 1000$.

Figure 4: Value (as defined in (14)) and log visitation count for each state in DeepSea under soft Q-learning. Darker color indicates larger value.

algorithm with a bound on regret which matches the best-known regret bound for Thompson sampling up to log factors and is close to the known lower bound. We call the algorithm 'K-learning', since the 'K-values' capture something about the epistemic *knowledge* that the agent can obtain by visiting each state-action. In the limit of zero uncertainty the K-values reduce to the optimal Q-values.

Although K-learning and Thompson sampling have similar theoretical and empirical performance, K-learning has some advantages. For one, it was recently shown that K-learning extends naturally to the case of two-player games and continues to enjoy a sub-linear regret bound, whereas Thompson sampling can suffer linear regret [34]. Secondly, Thompson sampling requires sampling from the posterior over MDPs and solving the sampled MDP exactly at each episode. This means that Thompson sampling does not have a 'fixed' policy at any given episode since it is determined by the posterior information plus a sampling procedure. This is typically a deterministic policy, and it can vary significantly from episode to episode. By contrast K-learning has a single fixed policy at each episode, the Boltzmann distribution over the K-values, and it is determined entirely from the posterior information (*i.e.*, no sampling). Moreover, K-learning requires solving a Bellman equation that changes slowly as more data is accumulated, so the optimal K-values at one episode are close to the optimal K-values for the next episode, and similarly for the policies. This suggests the possibility of approximating K-learning in an online manner and making use of modern deep RL

techniques such as representing the K-values using a deep neural network [29]. This is in line with the purpose of this paper, which is not to get the best theoretical regret bound, but instead to derive an algorithm that is close to something that is practical to implement in a real RL setting. Soft-max updates, maximum-entropy RL, and related algorithms are very popular in deep RL. However, they are not typically well motivated and they cannot perform deep-exploration. This paper tackles both those problems since the soft-max update, entropy regularization, and deep-exploration all fall out naturally from a utility maximization point of view. The popularity and success of these other approaches, despite their evident shortcomings in data efficiency, suggest that incorporating changes derived from K-learning could yield big performance improvements in real-world settings. We leave exploring that avenue to future work.

## Acknowledgments and Disclosure of Funding

I would like to thank Ian Osband, Remi Munos, Vlad Mnih, Pedro Ortega, Sebastien Bubeck, Csaba Szepesvári, and Yee Whye Teh for valuable discussions and clear insights. I received no specific funding for this work.

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
