# A Appendix

This appendix is dedicated to proving Theorem 1. First, we introduce some notation. The cumulant generating function of random variable $X : \Omega \to \mathbb{R}$ is given by

$$G^X(\beta) = \log \mathbb{E} \exp(\beta X).$$

We shall denote the cumulant generating function of $\mu_l(s, a)$ at time $t$ as $G_l^{\mu|t}(s, a, \cdot)$ and similarly the cumulant generating function of $Q_l^\star(s, a)$ at time $t$ as $G_l^{Q|t}(s, a, \cdot)$, specifically

$$G_l^{\mu|t}(s, a, \beta) = \log \mathbb{E}^t \exp(\beta \mu_l(s, a)), \quad G_l^{Q|t}(s, a, \beta) = \log \mathbb{E}^t \exp(\beta Q_l^\star(s, a)),$$

for $l = 1, \ldots, L$.

**Theorem 1.** *Under assumption 1 the K-learning algorithm 1 satisfies Bayes regret bound*

$$\begin{aligned}
\mathcal{BR}_\phi(T) &\leq 2\sqrt{(\sigma^2 + L^2)LSAT \log A(1 + \log T/L)} \\
&= \tilde{O}(L^{3/2}\sqrt{SAT}).
\end{aligned} \tag{15}$$

*Proof.* Lemma 1 tells us that $G^{Q|t}(s, a, \beta)$ satisfies a Bellman inequality for any $\beta \geq 0$. This implies that for fixed $\tau_t \geq 0$ the certainty equivalent values $\mathcal{Q}_l^t = \tau_t G_l^{Q|t}(s, a, 1/\tau_t)$ satisfy a Bellman inequality with optimistic Bellman operator $\mathcal{B}^t$ defined in Eq. (6), *i.e.*,

$$\mathcal{Q}_l^t \leq \mathcal{B}_l^t(\tau_t, \mathcal{Q}_{l+1}^t),$$

for $l = 1, \ldots, L$, where $\mathcal{Q}_{L+1}^t \equiv 0$. By construction the K-values $K^t$ are the unique fixed point of the optimistic Bellman operator. That is, $K^t$ has $K_{L+1}^t \equiv 0$ and

$$K_l^t = \mathcal{B}_l^t(\tau_t, K_{l+1}^t) \tag{16}$$

for $l = 1, \ldots, L$. Since log-sum-exp is nondecreasing it implies that the operator $\mathcal{B}_l^t(\tau, \cdot)$ is nondecreasing for any $\tau \geq 0$, *i.e.*, if $x \geq y$ pointwise then $\mathcal{B}_l^t(\tau, x) \geq \mathcal{B}_l^t(\tau, y)$ pointwise for each $l$. Now assume that for some $l$ we have $K_{l+1}^t \geq \mathcal{Q}_{l+1}^t$, then

$$K_l^t = \mathcal{B}_l^t(\tau_t, K_{l+1}^t) \geq \mathcal{B}_l^t(\tau_t, \mathcal{Q}_{l+1}^t) \geq \mathcal{Q}_l^t,$$

and the base case holds since $K_{L+1}^t = \mathcal{Q}_{L+1}^t \equiv 0$. This fact, combined with Lemma 4 implies that

$$\mathbb{E}^t \max_a Q_l^\star(s, a) \leq \tau_t \log \sum_{a \in \mathcal{A}} \exp(\mathcal{Q}_l^t(s, a)/\tau_t) \leq \tau_t \log \sum_{a \in \mathcal{A}} \exp(K_l^t(s, a)/\tau_t) \tag{17}$$

since log-sum-exp is increasing and $\tau_t \geq 0$. The following variational identity yields the policy that the agent will follow:

$$\tau_t \log \sum_{a \in \mathcal{A}} \exp(K_l^t(s, a)/\tau_t) = \max_{\pi_l(s) \in \Delta_A} \left( \sum_{a \in \mathcal{A}} \pi_l(s, a) K_l^t(s, a) + \tau_t H(\pi_l(s)) \right)$$

for any state $s$, where $\Delta_A$ is the probability simplex of dimension $A - 1$ and $H$ denotes the entropy, *i.e.*, $H(\pi(s)) = -\sum_{a \in \mathcal{A}} \pi(s, a) \log \pi(s, a)$ [13]. The maximum is achieved by the policy

$$\pi_l^t(s, a) \propto \exp(K_l^t(s, a)/\tau_t).$$

This comes from taking the Legendre transform of negative entropy term (equivalently, log-sum-exp and negative entropy are convex conjugates [10, Example 3.25]). The fact that (9) achieves the maximum is readily verified by substitution.

Now we consider the Bayes regret of an agent following policy (9), starting from (4) we have

$$
\begin{aligned}
\mathcal{BR}_\phi(T) &\stackrel{(a)}{=} \mathbb{E}\sum_{t=1}^{N}\mathbb{E}_{s\sim\rho}\mathbb{E}^t(V_1^\star(s) - V_1^{\pi^t}(s)) \\
&= \mathbb{E}\sum_{t=1}^{N}\mathbb{E}_{s\sim\rho}\mathbb{E}^t\Big(\max_a Q_1^\star(s,a) - \sum_{a\in\mathcal{A}}\pi_1^t(s,a)Q_1^{\pi^t}(s,a)\Big) \\
&\stackrel{(b)}{\leq} \mathbb{E}\sum_{t=1}^{N}\mathbb{E}_{s\sim\rho}\Big(\tau_t\log\sum_{a\in\mathcal{A}}\exp(K_1^t(s,a)/\tau_t) - \sum_{a\in\mathcal{A}}\pi_1^t(s,a)\mathbb{E}^t Q_1^{\pi^t}(s,a)\Big) \\
&\stackrel{(c)}{\leq} \mathbb{E}\sum_{t=1}^{N}\mathbb{E}_{s\sim\rho}\Big(\sum_{a\in\mathcal{A}}\pi_1^t(s,a)\Big(K_1^t(s,a) - \mathbb{E}^t Q_1^{\pi^t}(s,a)\Big) + \tau_t H(\pi_1^t(s))\Big)
\end{aligned}
\tag{18}
$$

where (a) follows from the tower property of conditional expectation where the outer expectation is with respect to $\mathcal{F}_1, \mathcal{F}_2, \ldots$, (b) is due to (17) and the fact that $\pi^t$ is $\mathcal{F}_t$-measurable, and (c) is due to the fact that the policy the agent is following is the policy (9). If we denote by

$$
\Delta_l^t(s) = \sum_{a\in\mathcal{A}}\pi_l^t(s,a)\Big(K_l^t(s,a) - \mathbb{E}^t Q_l^{\pi^t}(s,a)\Big) + \tau_t H(\pi_l^t(s))
$$

then we can write the previous bound simply as

$$
\mathcal{BR}_\phi(T) \leq \mathbb{E}\sum_{t=1}^{N}\mathbb{E}_{s\sim\rho}\Delta_1^t(s).
$$

We can interpret $\Delta_l^t(s)$ as a bound on the expected regret in that episode when started at state $s$. Let us denote

$$
\tilde{G}_l^{\mu|t}(s,a,\beta) = G_l^{\mu|t}(s,a,\beta) + \frac{(L-l)^2\beta^2}{2(n_l^t(s,a)\vee 1)}.
$$

Now we shall show that for a fixed $\pi^t$ and $\tau_t \geq 0$ the quantity $\Delta^t$ satisfies the following Bellman recursion:

$$
\Delta_l^t(s) = \tau_t H(\pi_l^t(s)) + \sum_{a\in\mathcal{A}}\pi_l^t(s,a)\Big(\delta_l^t(s,a,\tau_t) + \sum_{s'\in\mathcal{S}}\mathbb{E}^t(P_l(s'\mid s,a))\Delta_{l+1}^t(s')\Big)
\tag{19}
$$

for $s\in\mathcal{S}$, $l = 1,\ldots,L$, and $\Delta_{L+1}^t \equiv 0$, where

$$
\delta_l^t(s,a,\tau) = \tau\tilde{G}_l^{\mu|t}(s,a,1/\tau) - \mathbb{E}^t\mu_l(s,a) \leq \frac{\sigma^2 + (L-l)^2}{2\tau(n_l^t(s,a)\vee 1)},
\tag{20}
$$

where the inequality follows from assumption 1 which allows us to bound $G_l^{\mu|t}$ as

$$
\tau G_l^{\mu|t}(s,a,1/\tau) \leq \mathbb{E}^t\mu_l(s,a) + \frac{\sigma^2}{2\tau(n_l^t(s,a)\vee 1)}
$$

for all $\tau \geq 0$. We have that

$$
\begin{aligned}
\mathbb{E}^t Q_l^{\pi^t}(s,a) &\stackrel{(a)}{=} \mathbb{E}^t\Big(\mu_l(s,a) + \sum_{s'\in\mathcal{S}}P_l(s'\mid s,a)V_{l+1}^{\pi^t}(s')\Big) \\
&\stackrel{(b)}{=} \mathbb{E}^t\mu_l(s,a) + \sum_{s'\in\mathcal{S}}\mathbb{E}^t P_l(s'\mid s,a)\mathbb{E}^t V_{l+1}^{\pi^t}(s') \\
&\stackrel{(c)}{=} \mathbb{E}^t\mu_l(s,a) + \sum_{s'\in\mathcal{S}}\mathbb{E}^t P_l(s'\mid s,a)\sum_{a'\in\mathcal{A}}\pi_{l+1}^t(s',a')\mathbb{E}^t Q_{l+1}^{\pi^t}(s',a'),
\end{aligned}
\tag{21}
$$

where (a) is the Bellman Eq. (2), (b) holds due to the fact that the transition function and the value function at the next state are conditionally independent, (c) holds since $\pi^t$ is $\mathcal{F}_t$ measurable.

Now we expand the definition of $\Delta^t$, using the Bellman equation that the K-values satisfy and Eq. (21) for the Q-values

$$\Delta_l^t(s) = \tau_t H(\pi_l^t(s)) + \sum_{a \in \mathcal{A}} \pi_l^t(s,a) \Big( \tau_t \tilde{G}_l^{\mu|t}(s,a,1/\tau_t) - \mathbb{E}^t \mu_l(s,a) +$$

$$\sum_{s' \in \mathcal{S}} \mathbb{E}^t P_l(s' \mid s,a) \big( \tau_t \log \sum_{a' \in \mathcal{A}} \exp K_{l+1}^t(s',a')/\tau_t - \sum_{a' \in \mathcal{A}} \pi_{l+1}^t(s',a') \mathbb{E}^t Q_{l+1}^{\pi^t}(s',a') \big) \Big)$$

$$= \tau_t H(\pi_l^t(s)) + \sum_{a \in \mathcal{A}} \pi_l^t(s,a) \Big( \delta_l^t(s,a,\tau_t) +$$

$$\sum_{s' \in \mathcal{S}} \mathbb{E}^t P_l(s' \mid s,a) \big( \tau_t H(\pi_{l+1}^t(s')) + \sum_{a' \in \mathcal{A}} \pi_{l+1}^t(s',a') \big( K_{l+1}^t(s',a') - \mathbb{E}^t Q_{l+1}^{\pi^t}(s',a') \big) \big) \Big)$$

$$= \tau_t H(\pi_l^t(s)) + \sum_{a \in \mathcal{A}} \pi_l^t(s,a) \Big( \delta_l^t(s,a,\tau_t) + \sum_{s' \in \mathcal{S}} \mathbb{E}^t P_l(s' \mid s,a) \Delta_{l+1}^t(s') \Big),$$

where we used the variational representation (8). We shall use this to 'unroll' $\Delta^t$ along the MDP, allowing us to write the regret upper bound using only *local* quantities.

An *occupancy measure* is the probability that the agent finds itself in state $s$ and takes action $a$. Let $\lambda_l^t(s,a)$ be the expected occupancy measure for state $s$ and action $a$ under the policy $\pi^t$ at time $t$, that is $\lambda_1^t(s,a) = \pi_1^t(s,a)\rho(s)$, and then it satisfies the forward recursion

$$\lambda_{l+1}^t(s',a') = \pi_l^t(s',a') \sum_{(s,a)} \mathbb{E}^t(P_l(s' \mid s,a)) \lambda_l^t(s,a),$$

for $l = 1,\ldots,L$, and note that $\sum_{(s,a)} \lambda_l^t(s,a) = 1$ for each $l$ and so it is a valid probability distribution over $\mathcal{S} \times \mathcal{A}$. Now let us define the following function

$$\Phi^t(\tau,\lambda) = \sum_{l=1}^{L} \sum_{(s,a)} \lambda_l^t(s,a) \left( \tau H \left( \frac{\lambda_l^t(s)}{\sum_b \lambda_l^t(s,b)} \right) + \delta_l^t(s,a,\tau) \right). \tag{22}$$

where $\lambda^t(s)$ is the vector corresponding to the occupancy measure values at state $s$. One can see that by unrolling the definition of $\Delta^t$ in (19) we have that

$$\mathbb{E}_{s \sim \rho} \Delta_l^t(s) = \Phi^t(\tau_t, \lambda^t).$$

In order to prove the Bayes regret bound, we must bound this $\Phi^t$ function. For the case of $\tau_t$ annealed according to the schedule of (10) and the associated expected occupancy measure $\lambda^t$ we do this using lemma 3. For the case of $\tau_t^\star$ the solution to (11) and the associated expected occupancy measure $\lambda^{t\star}$ lemma 5 proves that

$$\Phi^t(\tau_t^\star, \lambda^{t\star}) \leq \Phi^t(\tau_t, \lambda^t),$$

and so it satisfies the same regret bound as the annealed parameter. This result concludes the proof. $\square$

## A.1 Proof of Bellman inequality lemma 1

**Lemma 1.** *The cumulant generating function of the posterior for the optimal Q-values satisfies the following Bellman inequality for all $\beta \geq 0$, $l = 1,\ldots,L$:*

$$G_l^{Q|t}(s,a,\beta) \leq \tilde{G}_l^{\mu|t}(s,a,\beta) + \sum_{s' \in \mathcal{S}} \mathbb{E}^t P_l(s' \mid s,a) \log \sum_{a' \in \mathcal{A}} \exp G_{l+1}^{Q|t}(s',a',\beta).$$

*where*

$$\tilde{G}_l^{\mu|t}(s,a,\beta) = G_l^{\mu|t}(s,a,\beta) + \frac{(L-l)^2\beta^2}{2(n_l^t(s,a) \vee 1)}.$$

*Proof.* We begin by applying the definition of the cumulant generating function

$$
\begin{aligned}
G_l^{Q|t}(s,a,\beta) &= \log \mathbb{E}^t \exp \beta Q_l^\star(s,a) \\
&= \log \mathbb{E}^t \exp \Big( \beta \mu_l(s,a) + \beta \sum_{s' \in \mathcal{S}} P_l(s' \mid s,a) V_{l+1}^\star(s') \Big) \\
&= G_l^{\mu|t}(s,a,\beta) + \log \mathbb{E}^t \exp \Big( \beta \sum_{s' \in \mathcal{S}} P_l(s' \mid s,a) V_{l+1}^\star(s') \Big)
\end{aligned}
\tag{23}
$$

where $G_l^{\mu|t}$ is the cumulant generating function for $\mu$, and where the first equality is the Bellman equation for $Q^\star$, and the second one follows the fact that $\mu_l(s,a)$ is conditionally independent of downstream quantities. Now we must deal with the second term in the above expression.

Assumption 1 says that the prior over the transition function $P_l(\,\cdot\mid s,a)$ is Dirichlet, so let us denote the parameter of the Dirichlet distribution $\alpha_l^0(s,a) \in \mathbb{R}_+^S$ for each $(s,a)$, and we make the additional mild assumption that $\sum_{s' \in \mathcal{S}} \alpha_l^0(s,a,s') \geq 1$, *i.e.*, we start with a total pseudo-count of at least one for every state-action. Since the likelihood for the transition function is a Categorical distribution, conjugacy of the categorical and Dirichlet distributions implies that the posterior over $P_l(\,\cdot\mid s,a)$ at time $t$ is Dirichlet with parameter $\alpha_l^t(s,a)$, where

$$
\alpha_l^t(s,a,s') = \alpha_l^0(s,a,s') + n_l^t(s,a,s')
$$

for each $s' \in \mathcal{S}_{l+1}$, where $n_l^t(s,a,s') \in \mathbb{N}$ is the number of times the agent has been in state $s$, taken action $a$, and transitioned to state $s'$ at timestep $l$, and note that $\sum_{s' \in \mathcal{S}_{l+1}} n_l^t(s,a,s') = n_l^t(s,a)$, the total visit count to $(s,a)$.

Our analysis will make use of the following definition and associated lemma from [40]. Let $X$ and $Y$ be random variables, we say that $X$ is stochastically optimistic for $Y$, written $X \geq_{SO} Y$, if $\mathbb{E}u(X) \geq \mathbb{E}u(Y)$ for any convex increasing function $u$. Stochastic optimism is closely related to the more familiar concept of second-order stochastic dominance, in that $X$ is stochastically optimistic for $Y$ if and only if $-Y$ second-order stochastically dominates $-X$ [19]. We use this definition in the next lemma.

**Lemma 2.** *Let $Y = \sum_{i=1}^n A_i b_i$ for fixed $b \in \mathbb{R}^n$ and random variable $A$, where $A$ is Dirichlet with parameter $\alpha \in \mathbb{R}^n$, and let $X \sim \mathcal{N}(\mu_X, \sigma_X^2)$ with $\mu_X \geq \frac{\sum_i \alpha_i b_i}{\sum_i \alpha_i}$ and $\sigma_X^2 \geq (\sum_i \alpha_i)^{-1} \mathrm{Span}(b)^2$, where $\mathrm{Span(b)} = \max_i b_i - \min_j b_j$, then $X \geq_{SO} Y$.*

For the proof see [40]. In our case, in the notation of the lemma 2, $A$ will represent the transition function probabilities, and $b$ will represent the optimal values of the next state, *i.e.*, for a given $(s,a) \in \mathcal{S} \times \mathcal{A}$ let $X_t$ be a random variable distributed $\mathcal{N}(\mu_{X_t}, \sigma_{X_t}^2)$ where

$$
\mu_{X_t} = \sum_{s' \in \mathcal{S}} \Big( \alpha_l^t(s,a,s') V_{l+1}^\star(s') / \sum_x \alpha_l^t(s,a,x) \Big) = \sum_{s' \in \mathcal{S}} \mathbb{E}^t(P_l(s' \mid s,a)) V_{l+1}^\star(s')
$$

due to the Dirichlet assumption 1. Due to assumption 1 we know that $\mathrm{Span}(V_l^\star(s)) \leq L - l$, so we choose $\sigma_{X_t}^2 = (L-l)^2 / (n_l^t(s,a) \vee 1)$. Let $\mathcal{F}_t^V = \mathcal{F}_t \cup \sigma(V^\star)$ denote the union of $\mathcal{F}_t$ and the sigma-algebra generated by $V^\star$. Applying lemma 2 and the tower property of conditional expectation we have that for $\beta \geq 0$

$$
\begin{aligned}
\mathbb{E}^t \exp \Big( \beta \sum_{s' \in \mathcal{S}} P_l(s' \mid s,a) V_{l+1}^\star(s') \Big) &= \mathbb{E}_{V_{l+1}^\star} \Big( \mathbb{E}_P \Big( \exp \beta \Big( \sum_{s' \in \mathcal{S}} P_l(s' \mid s,a) V_{l+1}^\star(s') \Big) \Big| \mathcal{F}_t^V \Big) \Big| \mathcal{F}_t \Big) \\
&\leq \mathbb{E}_{V_{l+1}^\star} \Big( \mathbb{E}_{X_t} (\exp \beta X_t | \mathcal{F}_t^V) \Big| \mathcal{F}_t \Big) \\
&= \mathbb{E}_{V_{l+1}^\star} \Big( \exp(\mu_{X_t} \beta + \sigma_{X_t}^2 \beta^2 / 2) \Big| \mathcal{F}_t \Big) \\
&= \mathbb{E}_{V_{l+1}^\star}^t \exp \Big( \beta \sum_{s' \in \mathcal{S}} \mathbb{E}^t P_l(s' \mid s,a) V_{l+1}^\star(s') + \sigma_{X_t}^2 \beta^2 / 2 \Big),
\end{aligned}
\tag{24}
$$

the first equality is the tower property of conditional expectation, the inequality comes from the fact that $P_l(s' \mid s,a)$ is conditionally independent of $V_{l+1}^\star(s')$ and applying lemma 2, the next equality

is applying the moment generating function for the Gaussian distribution and the final equality is substituting in for $\mu_{X_t}$. Now applying this result to the last term in (23)

$$\log \mathbb{E}^t \exp \left( \beta \sum_{s' \in \mathcal{S}} P_l(s' \mid s, a) V_{l+1}^\star(s') \right)$$

$$\overset{(a)}{\leq} \log \mathbb{E}_{V_{l+1}^\star}^t \exp \left( \beta \sum_{s' \in \mathcal{S}} \mathbb{E}^t P_l(s' \mid s, a) V_{l+1}^\star(s') + \sigma_{X_t}^2 \beta^2 / 2 \right)$$

$$\overset{(b)}{=} \log \mathbb{E}_{Q_{l+1}^\star}^t \exp \left( \beta \sum_{s' \in \mathcal{S}} \mathbb{E}^t P_l(s' \mid s, a) \max_{a'} Q_{l+1}^\star(s', a') \right) + \sigma_{X_t}^2 \beta^2 / 2$$

$$\overset{(c)}{\leq} \sum_{s' \in \mathcal{S}} \mathbb{E}^t P_l(s' \mid s, a) \log \mathbb{E}_{Q_{l+1}^\star}^t \exp \left( \beta \max_{a'} Q_{l+1}^\star(s', a') \right) + \sigma_{X_t}^2 \beta^2 / 2$$

$$\overset{(d)}{\leq} \sum_{s' \in \mathcal{S}} \mathbb{E}^t P_l(s' \mid s, a) \log \sum_{a' \in \mathcal{A}} \exp G_{l+1}^{Q|t}(s', a')(\beta) + \frac{\beta^2 (L - l)^2}{2(n_l^t(s, a) \vee 1)}$$

where (a) follows from Eq. (24)) and the fact that log is increasing, (b) is replacing $V^\star$ with $Q^\star$, (c) uses Jensen's inequality and the fact that $\log \mathbb{E} \exp(\cdot)$ is convex, and (d) follows by substituting in for $\sigma_{X_t}$ and since the max of a collection of positive numbers is less than the sum. Combining this and (23) the inequality immediately follows. $\qquad \square$

### A.2   Proof of lemma 3

**Lemma 3.** *Following the policy induced by expected occupancy measure $\lambda_l^t \in [0, 1]^{S \times A}$, $l = 1, \ldots, L$, and the temperature schedule $\tau_t$ in (10) we have*

$$\mathbb{E} \sum_{t=1}^N \Phi^t(\tau_t, \lambda^t) \leq 2\sqrt{(\sigma^2 + L^2)LSAT \log A(1 + \log T/L)}.$$

*Proof.* Starting from the definition of $\Phi$

$$\Phi^t(\tau_t, \lambda^t) = \sum_{l=1}^L \sum_{(s,a)} \lambda_l^t(s, a) \left( \tau_t H \left( \frac{\lambda_l^t(s)}{\sum_b \lambda_l^t(s, b)} \right) + \delta^t(s, a, \tau_t) \right)$$

$$\leq \tau_t L \log A + \tau_t^{-1} \sum_{l=1}^L \sum_{(s,a)} \lambda^t(s, a) \frac{(\sigma^2 + L^2)}{2(n_l^t(s, a) \vee 1)}$$

which comes from the sub-Gaussian assumption on $G_l^{\mu|t}$ and the fact that entropy satisfies $H(\pi(\lambda_s)) \leq \log A$ for all $s$. These two terms summed up to $N$ determine our regret bound, and we shall bound each one independently. To bound the first term:

$$L \log A \sum_{t=1}^N \tau_t \leq (1/2) L \sqrt{(\sigma^2 + L^2) SA \log A(1 + \log T/L)} \sum_{t=1}^N 1/\sqrt{t}$$

$$\leq \sqrt{(\sigma^2 + L^2) LSAT \log A(1 + \log T/L)},$$

since $\sum_{t=1}^N 1/\sqrt{t} \leq \int_{t=0}^N 1/\sqrt{t} = 2\sqrt{N}$, and recall that $N = \lceil T/L \rceil$. For simplicity we shall take $T = NL$, *i.e.*, we are measuring regret at episode boundaries; this only changes whether or not there is a small fractional episode term in the regret bound or not.

To bound the second term we shall use the pigeonhole principle lemma 6, which requires knowledge of the process that generates the counts at each timestep, which is access to the *true* occupancy measure in our case. The quantity $\lambda^t$ is not the true occupancy measure at time $t$, which we shall denote by $\nu^t$, since that depends on $P$ which we don't have access to (we only have a posterior distribution over it). However it is the *expected* occupancy measure conditioned on $\mathcal{F}_t$, *i.e.*, $\lambda^t = \mathbb{E}^t \nu^t$, which is easily seen by starting from $\lambda_1^t(s, a) = \pi_1^t(s, a)\rho(s) = \nu_1^t(s, a)$, and then inductively

using:

$$\mathbb{E}^t \nu_{l+1}^t(s',a') = \mathbb{E}^t \Big( \pi_{l+1}^t(s',a') \sum_{(s,a)} P_l(s' \mid s,a) \nu_l^t(s,a) \Big)$$

$$= \pi_{l+1}^t(s',a') \sum_{(s,a)} \mathbb{E}^t(P_l(s' \mid s,a)) \mathbb{E}^t \nu_l^t(s,a)$$

$$= \pi_{l+1}^t(s',a') \sum_{(s,a)} \mathbb{E}^t(P_l(s' \mid s,a)) \lambda_l^t(s,a)$$

$$= \lambda_{l+1}^t(s',a')$$

for $l = 1, \ldots, L$, where we used the fact that $\pi^t$ is $\mathcal{F}_t$-measurable and the fact that $\nu_l(s,a)$ is independent of downstream quantities. Now applying lemma 6

$$\mathbb{E} \sum_{t=1}^N \sum_{(s,a)} \frac{\lambda_l^t(s,a)}{n_l^t(s,a)+1} = \mathbb{E} \sum_{t=1}^N \mathbb{E}^t \left( \sum_{(s,a)} \frac{\lambda_l^t(s,a)}{n_l^t(s,a)+1} \right)$$

$$= \mathbb{E} \left( \sum_{t=1}^N \sum_{(s,a)} \frac{\nu_l^t(s,a)}{n_l^t(s,a)+1} \right)$$

$$\leq AS(1 + \log N),$$

which follows from the tower property of conditional expectation and since the counts at time $t$ are $\mathcal{F}_t$-measurable. From Eq. (10) we know that sequence $\tau_t^{-1}$ is increasing, so we can bound the second term as

$$\mathbb{E} \sum_{t=1}^N \tau_t^{-1} \sum_{l=1}^L \sum_{(s,a)} \frac{\lambda_l^t(s,a)(\sigma^2 + L^2)}{2(n_l^t(s,a)+1)} \leq (1/2)(\sigma^2 + L^2)\tau_N^{-1} \mathbb{E} \sum_{l=1}^L \left( \sum_{t=1}^N \sum_{s,a} \frac{\lambda_l^t(s,a)}{n_l^t(s,a)+1} \right)$$

$$\leq (1/2)(\sigma^2 + L^2)\tau_N^{-1} \sum_{l=1}^L SA(1 + \log N)$$

$$= (1/2)(\sigma^2 + L^2)\tau_N^{-1} LSA(1 + \log N)$$

$$= \sqrt{(\sigma^2 + L^2)LSAT \log A(1 + \log T/L)}.$$

Combining these two bounds we get our result. $\qquad \square$

### A.3 Proof of maximal inequality lemma 4

**Lemma 4.** *Let $X_i : \Omega \to \mathbb{R}$, $i = 1, \ldots, n$ be random variables with cumulant generating functions $G^{X_i} : \mathbb{R} \to \mathbb{R}$, then for any $\tau \geq 0$*

$$\mathbb{E} \max_i X_i \leq \tau \log \sum_{i=1}^n \exp G^{X_i}(1/\tau). \tag{25}$$

*Proof.* Using Jensen's inequality

$$\mathbb{E} \max_i X_i = \tau \log \exp(\mathbb{E} \max_i X_i/\tau)$$

$$\leq \tau \log \mathbb{E} \max_i (\exp X_i/\tau)$$

$$\leq \tau \log \sum_{i=1}^n \mathbb{E} \exp X_i/\tau \tag{26}$$

$$= \tau \log \sum_{i=1}^n \exp G^{X_i}(1/\tau),$$

where the inequality comes from the fact that the max over a collection of nonnegative values is less than the sum. $\qquad \square$

## A.4 Derivation of dual to problem (11)

Here we shall the derive the dual problem to the convex optimization problem (11), which will be necessary to prove a regret bound for the case where we choose $\tau_t^\star$ as the temperature parameter. Recall that the primal problem is

$$
\begin{aligned}
\text{minimize} \quad & \mathbb{E}_{s\sim\rho}(\tau \log \sum_{a\in\mathcal{A}} \exp(K_1(s,a)/\tau)) \\
\text{subject to} \quad & K_l \geq \mathcal{B}_l^t(\tau, K_{l+1}), \quad l = 1, \ldots, L, \\
& K_{L+1} \equiv 0,
\end{aligned}
$$

in variables $\tau \geq 0$ and $K_l \in \mathbb{R}^{S\times A}$, $l = 1, \ldots, L$. We shall repeatedly use the variational representation of log-sum-exp terms as in Eq. (8). We introduce dual variable $\lambda \geq 0$ for each of the $L$ Bellman inequality constraints which yields Lagrangian

$$
\sum_{s\in\mathcal{S}} \rho(s) \sum_{a\in\mathcal{A}} (\pi_1(s,a)K_1(s,a) + \tau H(\pi_1(s))) + \sum_{l=1}^{L} \lambda^T(\mathcal{B}_l^t(\tau, K_{l+1}) - K_l).
$$

For each of the $L$ constraint terms we can expand the $\mathcal{B}_l$ operator and use the variational representation for log-sum-exp to get

$$
\sum_{(s,a)} \lambda_l(s,a)\Big(\tau\tilde{G}_l^{\mu|t}(s,a,1/\tau) + \sum_{s'\in\mathcal{S}} \mathbb{E}^t P_l(s'\mid s,a)\Big(\sum_{a'\in\mathcal{A}} \pi_{l+1}(s',a')K_{l+1}(s',a') + \tau H(\pi_{l+1}(s'))\Big) - K_l(s,a)\Big).
$$

At this point the Lagrangian can be expressed:

$$
\mathcal{L}(\tau, K, \lambda, \pi) = \sum_{s\in\mathcal{S}} \rho(s)\Big(\sum_{a\in\mathcal{A}}(\pi_1(s,a)K_1(s,a)) + \tau H(\pi_1(s))\Big) + \sum_{l=1}^{L}\sum_{(s,a)} \lambda_l(s,a)\Big(\tau\tilde{G}_l^{\mu|t}(s,a,1/\tau) +
$$
$$
+ \sum_{s'\in\mathcal{S}} \mathbb{E}^t P_l(s'\mid s,a)\Big(\sum_{a'\in\mathcal{A}} \pi_{l+1}(s',a')K_{l+1}(s',a') + \tau H(\pi_{l+1}(s'))\Big) - K_l(s,a)\Big).
$$

To obtain the dual we must minimize over $\tau$ and $K$. First, minimizing over $K_1(s,a)$ yields

$$
\rho(s)\pi_1(s,a) = \lambda_1(s,a)
$$

and note that since $\pi_1(s)$ is a probability distribution it implies that

$$
\sum_{a_1\in\mathcal{A}} \lambda_1(s,a) = \rho(s)
$$

for each $s \in \mathcal{S}_1$. Similarly we minimize over each $K_{l+1}(s',a')$ for $l = 1, \ldots, L$ yielding

$$
\lambda_{l+1}(s',a') = \pi_{l+1}(s',a') \sum_{(s,a)} \mathbb{E}^t P_l(s'\mid s,a)\lambda_l(s,a).
$$

which again implies

$$
\sum_{a'\in\mathcal{A}} \lambda_{l+1}(s',a') = \sum_{(s,a)} \mathbb{E}^t P_l(s'\mid s,a)\lambda_l(s,a).
$$

What remains of the Lagrangian is

$$
\sum_{l=1}^{L}\sum_{(s,a)} \lambda_l(s,a)\Big(\tau\tilde{G}_l^{\mu|t}(s,a,1/\tau) + \tau H(\pi_l(s))\Big)
$$

which, using the definition of $\delta$ in Eq. (20) can be rewritten

$$
\sum_{l=1}^{L}\sum_{(s,a)} \lambda_l(s,a)\mathbb{E}^t\mu_l(s,a) + \min_{\tau\geq 0} \sum_{l=1}^{L}\sum_{(s,a)} \lambda_l^t(s,a)\Big(\tau_t H\Big(\frac{\lambda_l(s)}{\sum_b \lambda_l(s,b)}\Big) + \delta_l^t(s,a,\tau_t)\Big).
$$

Finally, using the definition of $\Phi$ in (22) we obtain:

$$\text{maximize} \quad \sum_{l=1}^{L} \sum_{(s,a)} \lambda_l(s,a) \mathbb{E}^t \mu_l(s,a) + \min_{\tau \geq 0} \Phi^t(\tau, \lambda)$$

$$\text{subject to} \quad \sum_{a' \in \mathcal{A}} \lambda_{l+1}(s', a') = \sum_{(s,a)} \mathbb{E}^t(P_l(s' \mid s, a)) \lambda_l(s,a), \quad s' \in \mathcal{S}_{l+1}, \ l = 1, \ldots, L \tag{27}$$

$$\sum_{a_1} \lambda_1(s,a) = \rho(s), \quad s \in \mathcal{S}_1$$

$$\lambda \geq 0.$$

## A.5 Proof of Lemma 5

**Lemma 5.** *Assuming strong duality holds for problem* (11)*, and denote the primal optimum at time* $t$ *by* $(\tau_t^\star, K_l^{t\star})$ *then the policy given by*

$$\pi_l^t(s,a) \propto \exp(K_l^{t\star}(s,a)/\tau_t^\star)$$

*satisfies the Bayes regret bound given in Theorem 1.*

*Proof.* The dual problem to (11) is derived above as Eq. (27). Denote by $\mathcal{L}^t$ the (partial) Lagrangian at time $t$:

$$\mathcal{L}^t(\tau, \lambda) = \sum_{l=1}^{L} \sum_{(s,a)} \lambda_l(s,a) \mathbb{E}^t \mu_l(s,a) + \Phi^t(\tau, \lambda).$$

Denote by $\lambda^{t\star}$ the dual optimal variables at time $t$. Note that the value $\mathcal{L}^t(\tau_t^\star, \lambda_l^{t\star})$ provides an upper bound on $\mathbb{E}^t \max_a Q_1^\star(s,a)$ due to strong duality. Furthermore we have that

$$\sum_{l=1}^{L} \sum_{(s,a)} \lambda_l^{t\star}(s,a) \mathbb{E}^t \mu_l(s,a) = \mathbb{E}_{s \sim \rho} \mathbb{E}^t V_1^{\pi^t}(s),$$

and so using (4) we can bound the regret of following the policy induced by $\lambda^{t\star}$ using

$$\mathcal{BR}_\phi(T) \leq \mathbb{E} \sum_{t=1}^{N} \left( \mathcal{L}^t(\tau_t^\star, \lambda^{t\star}) - \sum_{l=1}^{L} \sum_{(s,a)} \lambda_l^{t\star}(s,a) \mathbb{E}^t \mu_l(s,a) \right) = \mathbb{E} \sum_{t=1}^{N} \Phi^t(\tau_t^\star, \lambda^{t\star}). \tag{28}$$

Strong duality implies that the Lagrangian has a saddle-point at $\tau_t^\star, \lambda^{t\star}$

$$\mathcal{L}^t(\tau_t^\star, \lambda) \leq \mathcal{L}^t(\tau_t^\star, \lambda^{t\star}) \leq \mathcal{L}^t(\tau, \lambda^{t\star})$$

for all $\tau \geq 0$ and feasible $\lambda$, which immediately implies the following

$$\Phi^t(\tau_t^\star, \lambda^{t\star}) = \min_{\tau \geq 0} \Phi^t(\tau, \lambda^{t\star}). \tag{29}$$

Now let $\tau_t$ be the temperature schedule in (10), we have

$$\mathcal{BR}_\phi(T) \leq \mathbb{E} \sum_{t=1}^{N} \Phi^t(\tau_t^\star, \lambda^{t\star}) = \mathbb{E} \sum_{t=1}^{N} \min_{\tau \geq 0} \Phi^t(\tau, \lambda^{t\star}) \leq \mathbb{E} \sum_{t=1}^{N} \Phi^t(\tau_t, \lambda^{t\star}) \leq \tilde{O}(L\sqrt{LSAT}),$$

where the last inequality comes from applying lemma 3, which holds for any occupancy measure when the agent is following the corresponding policy. $\qquad \square$

## A.6 Proof of pigeonhole principle lemma 6

**Lemma 6.** *Consider a process that at each time $t$ selects a single index $a_t$ from $\{1, \ldots, m\}$ with probability $p_{a_t}^t$. Let $n_i^t$ denote the count of the number of times index $i$ has been selected up to time $t$. Then*

$$\sum_{t=1}^{N} \sum_{i=1}^{m} p_i^t / (n_i^t \vee 1) \leq m(1 + \log N).$$

*Proof.* This follows from a straightforward application of the pigeonhole principle,

$$\sum_{t=1}^{N}\sum_{i=1}^{m} p_i^t/(n_i^t \vee 1) = \sum_{t=1}^{N} \mathbb{E}_{a_t \sim p^t}(n_{a_t}^t \vee 1)^{-1}$$

$$= \mathbb{E}_{a_0 \sim p^0, \ldots, a_N \sim p^t} \sum_{t=1}^{N} (n_{a_t}^t \vee 1)^{-1}$$

$$= \mathbb{E}_{a_0 \sim p^0, \ldots, a_N \sim p^t} \sum_{i=1}^{m} \sum_{t=1}^{n_i^N \vee 1} 1/t$$

$$\leq \sum_{i=1}^{m}\sum_{t=1}^{N} 1/t$$

$$\leq m(1 + \log N),$$

where the last inequality follows since $\sum_{t=1}^{N} 1/t \leq 1 + \int_{t=1}^{N} 1/t = 1 + \log N$. $\square$

## B  Compute requirements

All experiments were run on a single 2017 MacBook Pro.