# OpenReview forum: "Variational Bayesian Reinforcement Learning with Regret Bounds"
_NeurIPS.cc/2021/Conference — NeurIPS 2021 Poster_

### Official Review · Reviewer_DyxF · 2021-07-09

**Rating:** 6
**Confidence:** 3

**Summary:**

The paper proposes K-learning, a reinforcement learning exploration heuristic with Bayesian regret bounds.
The authors adopt a Bayesian perspective under which long-term (Q-)values have epistemic uncertainty associated with them. They argue that exploration behaviour can be induced by maximising a particular (exponential) utility function of the resulting distributions. It is shown, that these utility scores obey a Bellman-like operator and can be computed using dynamic programming and an algorithm for episodic settings is proposed.
The authors link the optimal policy maximising the new objective to entropy regularised approaches, discuss the optimal selection of a temperature parameter governing risk-sensitivity and provide formal regret bounds under certain assumptions regarding the structure of the agents prior.
Finally, K-learning is compared empirically to a number of exploration baselines where it performs competitively.


**Ethical Concerns:**

-

**Limitations And Societal Impact:**

-

**Main Review:**

The development of principled exploration exploration strategies for reinforcement learning is an active area of research and as such this work concerns a highly relevant problem.
I hadn't seen the ``````  utility-based'' perspective on deriving policies from Q-value distributions previously and found it compelling and useful. It raises interesting question regarding the existence of other utilities that may induce behaviours of interest (e.g. pessimistic/risk-averse for safety critical domains etc.) and admit Bellman like updates.

The paper overall is well-structured and provides a good intuition for the presented approach.

A couple of points for improvements are listed below:

Firstly, I think the discussion of limitations could be expanded. In particularly, an analysis of the run-time complexity seems useful. My understanding is, that computation of K-values requires a full back-up across the entire state-action space and horizon which seems to be equivalent to the solution of a sampled MDP in Thompson sampling. As this was criticised as too costly in the introduction, it might indicate that K-Learning suffers from similar scalability issues.

Additionally, it would be valuable to know, how K-learning might relate to discounted formulations, continuous domains or belief structures that do not match the assumptions of the regret analysis -- all of which to my knowledge Thompson sampling can cope with.

The related work section might also need to be expanded to include approaches focussing on (Bayesian) exploration bonuses (BEB, VBRB, etc.). A number of these come with regret bounds and seem to have a similar structure to the K-scores, warranting further discussion.

Finally, I found the mathematical notation and derivation difficult to follow in a few places.
In particular, there are no brackets for expectations making it difficult to follow what they are computed of / with respect to.
The symbols $\rho$ (Eq. 4) and $\sigma$ (Eq. 5) are not defined as far as I can tell.




**Time Spent Reviewing:**

5

---

> ### Author Response · Authors · 2021-08-09
> **Response to DyxF**
>
> Thank you for your careful consideration and thoughtful feedback!
>
> “discussion of limitations could be expanded.”
>
> We have expanded in the text on limitations. In particular on the run-time, which includes solving a full MDP at each time-step as you say. We have also clarified in the text the point about this in comparison to Thompson sampling. Basically it is as follows: The way TS works is it samples a policy from the posterior at each episode, then acts with it. This means that TS does not have a “fixed” policy at any given episode since it is determined by the posterior information _plus_ a sampling procedure. This is typically a deterministic policy, and it can vary significantly from episode to episode. By contrast K-learning has a single fixed policy at each episode, the Boltzmann / Gibbs distribution over the K-values, and it is determined entirely by the posterior information (only actions are sampled, not the policy object itself). The point we were trying to make in the text is that since K-learning has a fixed (stochastic) policy, this could be approximated by an online method where the K-values are estimated from data and then the policy is formed by a Gibbs distribution, which is not possible for Thompson sampling because it doesn't have a fixed policy (and any averaging of the policies would destroy the regret guarantees of TS). This was not clear initially (as you point out) and so we have rewritten that part to clarify what is meant.
>
> “K-learning might relate to discounted formulations, continuous domains or belief structures that do not match the assumptions of the regret analysis”.
>
> We have expanded on the discussion about discounted formulations, continuous domains and MDPs that do not match the assumptions of the regret analysis. In short:
> 1) discounted formulations are not a problem
> 2) continuous domains may be more challenging and we have added discussion on this
> 3) if the MDP is misspecified then we pay a multiplicative factor in the regret related to the Radon-Nikodym derivative of the true prior with respect to the assumed prior (see “Learning to Optimize via Posterior Sampling”, Russo and Van Roy 2013, Section 3.1, for a discussion of this). In other words if the prior is misspecified we still have a $O(\sqrt{T})$ regret bound so long as the true prior is absolutely continuous with respect to the misspecified prior.
>
> “The related work section might also need to be expanded.”
>
> Thanks for the pointers on BEB and VBRB, I have added those and several other citations to the text with discussion and added BEB to the experiments section.
>
>
> “The symbols $\rho$ (Eq. 4) and $\sigma$ (Eq. 5) are not defined”
>
> Thanks for noting that $\sigma$ is not defined. It is the noise in the reward process (ie, how noisy $r(s,a) - \mu(s,a)$ is). We have corrected this omission in the current draft. We believe $\rho$ is defined before we use it as the initial state distribution in section 2. If we have missed somewhere please let us know!

---

> > ### Comment · Reviewer_DyxF · 2021-08-23
> > **Post-rebuttal comments**
> >
> > Thank you for the detailed responses. This addresses all my concerns.

---

### Official Review · Reviewer_AGCq · 2021-07-16

**Rating:** 4
**Confidence:** 4

**Summary:**

This paper gives variational Bayesian reinforcement learning thereotical results with regret bounds  and an algorithm.

**Limitations And Societal Impact:**

Yes.

**Main Review:**

The paper considers the exploration-exploitation trade-off in reinforcement learning and show that an agent endowed with an exponential epistemic-risk-seeking utility function explores efficiently, as measured by regret.

Several techniques are incorporated to get the Bayes regret.

I think the theoretical result is interesting. However, the experiments are not sufficient at the current stage.

a. The paper assumes the MDP is time-inhomogeneous (see assumption 1) and states that by an unrolling procedure the method can be used to scenarios unfitting this assumption. Please give experiments to show the performance in this case.

b. The experiments are simple with a deepsea environment. Could the method in the paper be used to more complex and real environments and be compared with more SOTA methods?

**Time Spent Reviewing:**

10

---

> ### Author Response · Authors · 2021-08-09
> **Reponse to AGCq**
>
> Thank you for your time and effort reviewing our paper!
>
> “The paper assumes the MDP is time-inhomogeneous”.
>
> We have substantially rewritten that part of the text to make it clearer what precisely the framework and assumptions are. Note that the framework we consider is exactly the same as, eg, “Is Q-learning Provably Efficient?” by Jin et al 2018, and many other papers. We will also add an unrolled “Riverswim” environment discussion to demonstrate this.
>
>
> “Could the method in the paper be used to more complex and real environments and be compared with more SOTA methods.”
>
> We do compare against UCBVI, which as we understand has a theoretical regret bound matching the information theoretic lower bound (up to log factors), so is SOTA, at least in theory. We also compare our algorithm to ten other tabular algorithms, which we feel is quite exhaustive. This is primarily a theory paper about efficient tabular reinforcement learning, so we compare to other tabular environments on tabular domains. The DeepSea environment is especially crafted to be a ‘hard exploration problem’ and algorithms that do not come with guarantees (like epsilon greedy and soft Q-learning) take a very long time to learn. In that sense it is a complex and challenging environment. Deep RL algorithms like DQN, A3C, MPO, PPO, TRPO etc, do not have the exploration guarantees to perform well in this domain and will also take a long time to learn. This is demonstrated well by the Bsuite paper (Osband et al 2019) which tests some of these algorithms on the deepsea environment and shows that they do not work well on this task. The best performing deep RL algorithm for exploration we know of is bootstrapped DQN, but that is essentially a deep network implementation/approximation of Thompson sampling or RLSVI, both of which we already compare against in the deepsea experiment.
>
> Ideally, we would like a deep RL version of K-learning that performed well on more complex environments like Atari or Starcraft. However, converting a model-based tabular Bayesian algorithm into a model-free deep RL algorithm is no easy task and would be a paper (or a few papers) on its own (consider how many changes were needed for DQN to work as compared to tabular Q-learning). This paper lays the algorithmic and theoretical ground work for future work to develop deep RL approximations to K-learning. Does that address your concern?

---

### Official Review · Reviewer_biBz · 2021-07-16

**Rating:** 6
**Confidence:** 4

**Summary:**

The author considers the setting of the MDP is sampled from a known prior $\phi$ rather than a fixed unknown one, which belongs to Bayesian reinforcement learning. Under this setting, the author firstly introduces an expected upper bound on the optimal Q-value $Q^{\star}_l(s, a)$, then introduces a new operator called optimistic Bellman operator. This operator combines the principle of using upper confidence bound to encourage exploration and the randomness influence of the transition kernel and rewards, providing a sequence of optimistic estimation of the optimal Q-value. Furthermore, it also uses a soft-max exploration policy to sample from the environment.

Based on these components, K-learning is proposed and has been proved to enjoy a tight Bayes regret bound $O(\sqrt{L^3SAT})$. More numerical experiments are conducted to show the performance with comparisons to existing methods.


**Limitations And Societal Impact:**

Note:
1. There is no definition for $\sigma$ in line 161 and later. Furthermore, since $\sigma$ appears in the regret bound in the main theorem 1, it is hard to tell whether $\sigma$ is a universal constant or is actually related to the important parameters $L, T, S, A$.

2. The assumptions in this paper seem only for the convenience of the proof, such as ensuring the Dirichlet prior to the transition in Assumption 4. It seems not reasonable in the real world? Can you justify these assumptions more carefully?
3. This is just a question that makes me confusing:
It seems the author utilizes a Hoeffding-style exploration bonus. However, researchers can only achieve $O(\sqrt{L^4SAT})$ regret bound in [1] using such bonus, while achieves $O(\sqrt{L^3SAT})$ using Bernstein-style exploration bonus. Is it owing to some assumptions in the paper or some other reason?



**Main Review:**

Originality:
In the Bayesian reinforcement learning setting (MDP is sampled from a known prior $\phi$), the algorithm K-learning is proposed to better balance exploration-exploitation by adding an exploration bonus and choosing the risk-seeking parameter (similar to the parameter in soft-Q-learning). Since the exploration bonus and the soft-Q-learning has already been proposed before, this work can be seen as proposing an algorithm using these principles in a new setting, which is of moderate originality.

Quality:
The results seem reasonable, while I didn't read all the proof in the appendix.

Clarify:
The paper is well-organized and is clear.

Significance:
1. It will be better to introduce more application scenarios that satisfy the Bayesian reinforcement learning, since the assumption of transition kernel is under some known prior is not well-explained in this work.
2. Although intuition is introduced in lines 169-179, line 191, etc, the technical difficulty or novelty is not mentioned enough.
3. The contributions of this work are not well-compared with existing works such as [1], [2], which also utilize exploration bonuses and even add more variance reduction techniques.




[1] Jin, C., Allen-Zhu, Z., Bubeck, S., and Jordan, M. I. (2018). Is Q-learning provably efficient? In Advances
in Neural Information Processing Systems, pages 4863–4873
[2] Zhang, Z., Zhou, Y., and Ji, X. (2020b). Almost optimal model-free reinforcement learning via reference- advantage decomposition. Advances in Neural Information Processing Systems, 33.
[3] Li, Gen, et al. "Sample complexity of asynchronous Q-learning: Sharper analysis and variance reduction." arXiv preprint arXiv:2006.03041 (2020).



**Time Spent Reviewing:**

3

---

> ### Author Response · Authors · 2021-08-09
> **Response to biBz**
>
> Thank you for your careful review and suggestions!
>
> “It will be better to introduce more application scenarios that satisfy the Bayesian reinforcement learning”
>
> We have expanded on this in the text (and see the discussion in the main response above). Note that the regret bound we prove holds for *any* prior (so long as it is known). Also recall that a $O(\sqrt{T})$ regret bound with respect to any prior implies that the regret bound for any family of MDPs under the prior must be $O(\sqrt{T})$ eventually so long as that instance has support under the prior (see appendix A in “(More) Efficient Reinforcement Learning via Posterior Sampling” Osband and Van Roy 2013 and Section 3.1 “Learning to Optimize via Posterior Sampling”, Russo and Van Roy 2013). Loosely speaking, this holds because if there was some set of problems under the prior had $O(T)$ regret, then they would eventually dominate the Bayesian regret bound and yield $O(T)$ Bayesian regret. Finally we note that Bayesian RL algorithms have the added advantage of being better able to exploit prior knowledge about the problem, and are often the best performing approaches in practice (see numerical results section).
>
> “Although intuition is introduced in lines 169-179, line 191, etc, the technical difficulty or novelty is not mentioned enough.”
>
> Good point on the technical difficulties not being discussed. We have added a paragraph where we discuss the proof-steps at a high level and where the difficulty in each step arises.
>
> “The contributions of this work are not well-compared with existing works such as [1], [2]...[3]”
>
> Thanks for pointing this out. We have added citations and discussion on those three papers. We have also added the Q-learning method of Jin et al. to the deepsea experiments to show precisely how well our method does against a model-free algorithm in practice.
>
> “no definition for $\sigma$“
>
> Thanks for noting that $\sigma$ is not defined. It is the noise in the reward process (ie, how noisy $r(s,a) - \mu(s,a)$ is). We have corrected this omission in the current draft.
>
> “It seems the author utilizes a Hoeffding-style exploration bonus. However, researchers can only achieve $O(\sqrt{L^4SAT}))$ regret bound in [1] using such bonus”.
>
> We believe there are a few sources of differences. For one, note that the Jin et al paper is model-free, whereas this work is model-based. This is a big departure and an advantage for the Jin et al paper, since it requires less storage. Secondly, we do not use Hoeffding bounds exactly, though they are superficially similar. We use bounds arising from the cumulant generating function of the posterior, which tend to be marginally less crude. A closer comparison to our work is PSRL (Osband and Van Roy 2016), which achieves similar regret bounds. We have added discussion on this where we reference the Jin et al. paper, including the fact that the Jin et al. paper has the advantage of being model-free.

---

> > ### Comment · Reviewer_biBz · 2021-08-18
> > **Reply to the author.**
> >
> > Thanks for addressing my problems. The author's response convinced me of most of the raised problems. However, the contribution compared with existing works is still not clearly mentioned. What's the advantage of this work compared to the model-free algorithms in [1][2][3] and the model-based work UCB-VI in [4]{Azar, Mohammad Gheshlaghi, Ian Osband, and Rémi Munos. "Minimax regret bounds for reinforcement learning." International Conference on Machine Learning. PMLR, 2017.} Since the results in [2] [3]and [4] already achieved the optimal regret $O(\sqrt{L^2SAT})$ as long as the episode size exceeds a warm-up sample complexity, without knowing any prior knowledge.

---

> > > ### Author Response · Authors · 2021-08-20
> > > **Reply to reviewer biBz**
> > >
> > > Yes, good question. We need to add a discussion on this to the related work section. At a high level there are some advantages to K-learning over these other approaches. First, K-learning is Bayesian so can better incorporate prior knowledge. Second, in practice K-learning often out-performs these other algs (see the numerical results section for K-learning vs UCB-VI. We have also added optimistic Q-learning by Jin et al. [1] to the latest results which performs worse than UCB-VI). Unlike these other algorithms, K-learning (at least the variant that uses the optimal temperature) is _stationary_, in that it doesn't depend on time directly, only the posteriors. Unlike Thompson sampling, the K-learning policy is a deterministic function of the posterior beliefs (TS samples), so K-learning is more amenable to an online approximation.
> > >
> > > Finally, note that the purpose of this paper is not to get the best regret bound, but instead to derive an algorithm that is close to something that is practical to implement in a real RL setting (eg, in deep RL). Soft-Q-learning, max-entropy RL and related algorithms are very popular in deep RL. However, they are not well motivated and they cannot perform deep-exploration. This paper tackles both those problems - the soft update, the entropy regularization, and the deep-exploration all fall out naturally from a utility maximization point of view. The hope is that this paper will allow practitioners to improve their deep RL soft-Q or max-ent implementations so as to make them more data efficient, as well as to increase the understanding about how those methods can be motivated / derived. It is true that K-learning is model-based, but the objective is to start in this paper with the machinery and understanding we developed and then try to derive a model-free algorithm based on K-learning with similar guarantees in future work.

---

> > > > ### Comment · Reviewer_biBz · 2021-08-20
> > > > **Response to the author.**
> > > >
> > > > Thank you for answering this question in detail. The answer already solves all my concerns.

---

### Official Review · Reviewer_5rQg · 2021-07-21

**Rating:** 6
**Confidence:** 3

**Summary:**

This paper studies the problem of Bayesian RL. The proposed algorithm is based on a soft-Q backward propagation and achieves regret bound of $\tilde{O}(L\sqrt{TSA})$.

**Ethical Concerns:**

There is no ethical issues.

**Ethics Review Area:**

["I don’t know"]

**Limitations And Societal Impact:**

Yes.

**Main Review:**

Overall the paper is well written and the proofs seem correct for me. The authors show that soft-Q backward propagation is powerful to guarantee good regret performance by tuning the temperature parameter properly.

Minor comments:
1. It might be incorrect to claim that the $\tilde{O}(L\sqrt{SAT})$ regret bound is tight. In finite-horizon case, the general minimax regret lower bound is $\Omega(\sqrt{L^3XAK})$, where $K=T/L$ is the number of episodes and $X=S/L$ is the number of states per layer. There is still a $\sqrt{L}$ gap between the proposed upper bound and lower bound.

2. The second way to find $\tau_{t}$ is unnecessary since the final selection of $\tau_{t}$ could be simple.

3. eq(23): $=\to \leq $;

4. Line541 equation (c): it is better to highlight that the convexity of $\log(E[\exp(X)])$ is used.

**Time Spent Reviewing:**

6

---

> ### Author Response · Authors · 2021-08-09
> **Response to 5rQg**
>
> Thank you for your careful review, and for spotting typos deep in the appendix! These have now been fixed (the $\leq$ and the expansion on the convexity of $\log \mathbf{E} \exp$). In response to your more significant comments:
>
> “It might be incorrect to claim that the $\tilde O(L \sqrt{SAT})$ regret bound is tight.”
>
> On lines 49-51 the submitted draft says “This regret bound ... is within a factor of $\sqrt{L}$ of the known minimax lower bound of $\Omega(L \sqrt{ SAT})$ [19].” We don't claim that the K-learning bound is tight, indeed it is $\sqrt{L}$ away from the known lower bound. We believe that lower bound is correct for the setting we consider, where the rewards and transition function can be different for each time-period $l$ (see Appendix D of Jin et al 2018 “IS Q-learning provably efficient?”). Note that if we know that the reward and transition function are the _same_ for each time period (eg, $r_l(s,a)$ depends on $(s,a)$ but not on $l$) then we believe it shaves off a factor of $\sqrt{L}$ from the lower bound, which would match your lower bound (so it's just a different setting). I have cleaned up the preliminaries section a bit to make it clearer that the setup we consider in this paper is the same as the Jin et al paper. Does this align with your understanding?
>
> “The second way to find $\tau_t$ is unnecessary since the final selection of $\tau_t$ could be simple.”
>
> It is true that the second way to compute $\tau_t$ is unnecessary since the schedule is sufficient to produce the bound, however computing $\tau_t$ via the convex optimization problem tends to perform better in practice (though it is more computationally expensive). Some papers attempt to explicitly optimize this parameter (eg, “V-MPO: On-policy maximum a posteriori policy optimization for discrete and continuous control” Song et al.) and so providing a principled way to do this with a regret guarantee is useful to the community. We have expanded upon this in the text.

---

### Author Response · Authors · 2021-08-09
**Thank you to the reviewers**

We thank the reviewers for their careful reading of the paper and their insightful feedback. We respond to each reviewer below. We would be grateful if the reviewers could take a look at the responses and adjust the scores if we have addressed their concerns appropriately. If we have missed anything please let us know.

One point brought up by several reviewers is the missing definition of $\sigma$ - indeed this is missing and we apologize for the oversight! It is the noise in the reward process (ie, how noisy $r(s,a) - \mu(s,a)$ is). I have corrected this omission in the current draft (and fixed a few other typos).

One question a couple of reviewers raised is (paraphrased): “When or how is Bayesian RL appropriate?”. To begin with we should mention that Bayesian algorithms can incorporate prior knowledge and information easily, so if there is some nice structure in the problem K-learning (and related algorithms like Thompson sampling) are better able to exploit it than cruder upper confidence bounds approaches. They also tend to perform better in practice (see the numerical experiments section in the draft where the best performing algorithms are Bayesian despite this being a ‘fixed’ problem instance not drawn from a prior). Moreover, Bayesian regret bounds can be informative on a per-instance basis. Firstly, in order to achieve a Bayesian regret bound of $O( \sqrt{T} )$ then clearly the probability of the set of problems that have linear regret must decrease at least as fast as $O( 1/ \sqrt{T} )$ under the prior. In fact, we can make a slightly stronger argument about the frequentist regret for any algorithm that achieves $O( \sqrt{T} )$ Bayesian regret using Markov’s inequality, as made in Appendix A of Osband and Van Roy 2013: Let $\mathcal{M}$ be any family of MDPs with non-zero probability under the prior, and denote the true MDP by $M^\star$. Denote the _instance_ regret of the algorithm up to time $T$ following policy $\pi$ on MDP $M^\star$ as $\mathrm{Regret}(T, \pi, M^\star)$, where we assume that the algorithm that generates $\pi$ has _Bayesian_ regret bounded by
$O(\sqrt{T})$. Then for any $\epsilon > 0$ and $\alpha > 1/2$ they show that

$$
\mathbf{Prob}\left(\frac{\mathrm{Regret}(T, \pi, M^\star)}{T^\alpha} > \epsilon \biggm| M^\star \in \mathcal{M}\right) \rightarrow 0.
$$

In other words so long as $M^\star$ is in the support of the prior, then the
instance regret is asymptotically close to the $O(\sqrt{T})$ lower bound. We have greatly expanded on our discussion about this in the text.

Finally, based on the suggestions from the reviewers, we added several extra algorithms to the experiments and comparisons we ran: UCRL2, UCFH, BEB, BOLT, and Optimistic Q-learning (Jin et al 2018). The results of these additional experiments do not change the conclusion of the paper, because the two best performing algorithms are still K-learning and Thompson sampling.

---

### Author Response · Authors · 2021-08-29
**Thanks again**

We would like to thank the reviewers for their thorough feedback, for engaging in the rebuttal, and for their original evaluation of the paper. This is much appreciated and we believe that the paper has gained a lot from their feedback. Based on our discussions and the changes we will incorporate would any of the reviewers be willing to revise their scores? Or are there more concerns / questions the reviewers have that we can answer?

---

> ### Comment · Reviewer_5rQg · 2021-08-30
> **Keep my score**
>
> Thanks AC for leading discussion. The response addressed my concern and I'd like to keep my score as 6.

---

### Decision · Program_Chairs · 2021-09-27

**Decision:**

Accept (Poster)

**Comment:**

This paper proposes K-learning, a reinforcement learning exploration algorithm with Bayesian regret bounds. The paper is in a tabular setting that works for an interesting exponential epistemic-risk-seeking utility function.  Although the regret bounds may not be tight, most of the reviewers believe that the contribution is sufficient for acceptance. The experiment part is also a plus.